# High-Temporal-Resolution Forest Growth Monitoring Based on Segmented 3D Canopy Surface from UAV Aerial Photogrammetry

**Wenbo Zhang [1], Feng Gao [1], Nan Jiang [1], Chu Zhang [2] and Yanchao Zhang [1,*]**

1    School of Mechanical Engineering and Automation, Zhejiang Sci-Tech University, Hangzhou 310018, China;
     202020602086@mails.zstu.edu.cn (W.Z.); 2018330301115@mails.zstu.edu.cn (F.G.);
     2019330301158@mails.zstu.edu.cn (N.J.)
2    School of Information Engineering, Huzhou University, Huzhou 313000, China; chuzh@zjhu.edu.cn
*    Correspondence: yczhang@zstu.edu.cn

**Abstract:** Traditional forest monitoring has been mainly performed with images or orthoimages from aircraft or satellites. In recent years, the availability of high-resolution 3D data has made it possible to obtain accurate information on canopy size, which has made the topic of canopy 3D growth monitoring timely. In this paper, forest growth pattern was studied based on a canopy point cloud (PC) reconstructed from UAV aerial photogrammetry at a daily interval for a year. Growth curves were acquired based on the canopy 3D area (3DA) calculated from a triangulated 3D mesh. Methods for canopy coverage area (CA), forest coverage rate, and leaf area index (LAI) were proposed and tested. Three spectral vegetation indices, excess green index (ExG), a combination of green indices (COM), and an excess red union excess green index (ExGUExR) were used for the segmentation of trees. The results showed that (1) vegetation areas extracted by ExGUExR were more complete than those extracted by the other two indices; (2) logistic fitting of 3DA and CA yielded S-shaped growth curves, all with correlation $R^2 > 0.92$; (3) 3DA curves represented the growth pattern more accurately than CA curves. Measurement errors and applicability are discussed. In summary, the UAV aerial photogrammetry method was successfully used for daily monitoring and annual growth trend description.

**Keywords:** UAV; remote sensing; aerial photogrammetry; 3D area; SVIs; forest growth curve

## 1. Introduction

Forests represent the primary force for absorbing and storing carbon to fight global warming. It is increasingly understood that forests store large quantities of carbon in both vegetation and soil and exchange carbon with the atmosphere. The ability to mitigate the impacts of climate change through the enhancement of carbon sequestration has been identified [1]. Meanwhile, forests generate essential raw materials for a wide range of needs, from household uses, such as cooking fuel, flooring, furniture, and house construction, to industrial uses, such as wooden boats, decoration, etc. Moreover, forests have made a significant contribution to water conservation, wind and sand control, and biodiversity protection.

Research on forest monitoring and analysis plays a fundamental role in understanding growth patterns and provides information for policy development to improve forest management. Traditional forest monitoring research has been based mainly on remote satellite or aircraft images. The image resolution of these data is usually >1 m, and the temporal resolution is generally over 1 week [2,3]. For example, the latest Landsat 9 satellite had a temporal resolution of 16 days when it worked alone and 8 days when it worked with the Landsat 8 satellite [4]. This temporal resolution is not sufficient for a detailed study of forest growth. For example, forests grow very fast in spring and experience dramatic

canopy/structure/size change, and in autumn, forests may turn yellow/red overnight. High-temporal- and -spatial-resolution data are essential for monitoring forest dynamics and studying forest growth patterns in annual cycles [5,6]. Compared with satellites, drones with imaging sensor onboard have apparent advantages such as a short revisiting cycle, flexibility, and high image resolution, which have made UAVs an important emerging remote sensing platform for forest growth monitoring and evaluation [7,8]. Moreover, UAVs can still be used when the weather is cloudy, while satellites cannot deal with images when thick clouds exist [9,10]. Various sensors can be mounted on UAVs, such as thermal infrared cameras [11], hyperspectral cameras [12], and LiDAR [13]. UAVs have been used for forest biomass estimation [14], tree height measurement [15], pest and disease monitoring [16,17], disaster assessment [18–20], fire surveillance and prevention [21], etc. UAV remote sensing data have high image and temporal resolution. These features facilitate immediate monitoring of forest disturbances and timely response and damage assessment after a disaster. Despite the numerous pros of UAVs, some cons need to be pointed out. For example, small and light commercial drones are prohibited from flying over 120 m above the ground because of safety policies. In addition, since the weight of drones is strictly limited, their flight durations are short (20–30 min) [22]. Thus, the range and coverage of a single flight are much smaller than those of satellites or aircraft [23]. Usually, drones can hardly be used for city- or state-level monitoring.

Current forest growth studies are primarily performed with multispectral or RGB orthoimages using different spectral vegetation indices (SVI) for ground cover segmentation [24–26]. However, analysis based on images is a kind of planimetric analysis that suffers from dimensional loss. Spectral inversion-based vegetation indices, such as LAI [27] and the most commonly used normalized difference vegetation index (NDVI) [28], do not depict forest growth directly or visibly enough. Computer vision and airborne LiDAR provide new methods for forest growth monitoring that it possible to depict forest growth directly and visibly in three dimensions. Structure from motion (SfM) [29] is a frequently used aerial photogrammetry method for 3D surface reconstruction, and through it, 3D point cloud (PC) data can be obtained. It estimates the 3D structure from 2D images containing visual motion information. These images are spatially correlated. In [30], SfM photogrammetry was successfully applied in forestry research, showing that SfM had comparable resolution and precision to those of LiDAR in surface shape acquisition. The generated 3D PC and digital shape models have been successfully applied in tree height measurement [31], canopy volume and cover area calculation [32], tree species classification, and age statistics [33]. A generated PC was converted to canopy height models (CHM) for dynamics analysis. In [34], SfM was used to create an accurate forest structure and relevant spectra to measure tree height and canopy dynamics. There is a lack of forest monitoring research based on 3D canopy structures. In 3D canopy analysis, an index-based method that directly reflects the 3D dynamics of the forest is needed, such as the 3D area (3DA). In point cloud processing, the 3DA is calculated after a long process including segmentation, filtering, meshing, etc. Overall, in recent times the availability of high-resolution 3D data has made it possible to obtain accurate information on canopy size, which has made the topic of canopy 3D growth monitoring timely.

Therefore, the objectives of this research were: (1) to find out the growth curve of the forest in the experimental area at a daily interval. Forest growth was measured from the reconstructed canopy 3D surface area. Different SVI-based segmentations for 3D PCs reconstructed from UAV aerial photogrammetry were explored; (2) to study the growth curves of the forest canopy's 3D area (total unilateral area of the upper surface in the canopy) and coverage area (area of the canopy orthographic projection to the ground) and to compare the measurements and curves obtained by various SVIs; (3) to explore forest health estimation based on forest health status and ecosystem health status in the experimental area through the estimation of LAI and forest coverage rate (ratio of forest area to land area).

## 2. Materials and Methods

In Figure 1, the workflow of this study is illustrated. It can be divided into image acquisition, PC generation and processing, canopy area calculation, and growth curve fitting.

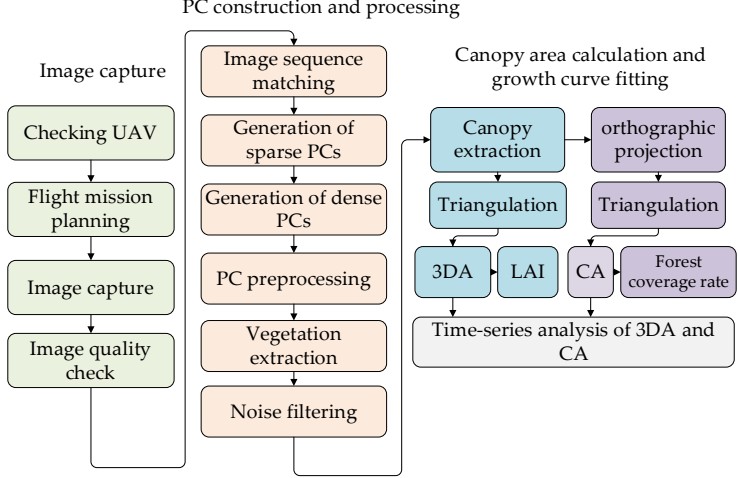

**Figure 1.** The workflow of this study. (1) Forest image acquisition based on UAV. (2) 3D PCs were reconstructed from these images, and noise points were removed (PCs preprocessing). (3) Points representing vegetation were extracted from the PCs, then the canopy was extracted from vegetation. (4) Triangulate the extracted canopy PC and calculate 3DA; make the canopy projection to the ground to obtain CA. (5) Time-series analysis of 3DA and CA measurements to explore the forest growth pattern.

### 2.1. Remote Sensing Image Capture and 3D PCs Reconstruction

2.1.1. Remote Sensing Image Capturing

A forest was chosen as the test subject. It was located at a campus in Hangzhou (30°32′N, 120°36′E, WGS 84), China, as shown in Figure 2a. The size of the experimental area was 107 m × 156 m. There were more than 30 species of trees in the experimental area, with deciduous trees covering 64% of the forest and evergreen trees covering 36%. This complex forest could test the generality of the proposed method.

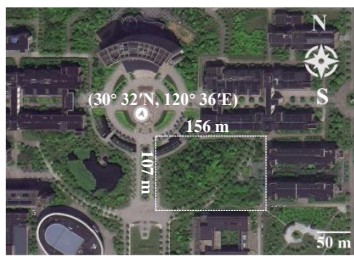 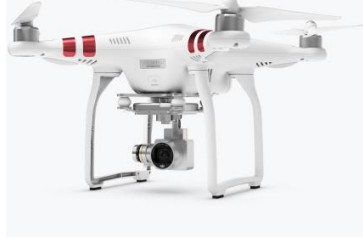 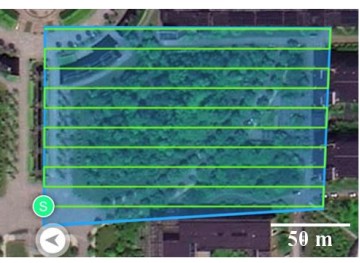

(**a**) Experimental area      (**b**) DJI Phantom 3 UAV      (**c**) UAV flight path

**Figure 2.** The experimental area and mission design. (**a**) The rectangular area marked by white lines represents the experimental area. (**b**) The DJI quad-rotor aerial photography UAV that was used in the experiment. (**c**) The flight path design of the UAV.

The experiment lasted ten months, and the forest images were captured from 1 September 2020 to 27 June 2021 so that a complete annual growth cycle was covered. The shooting interval was a day. In order to reduce shadows in forest images, all photo-shooting flights were undertaken between 12:00 and 14:00. A commercial UAV DJI Phantom 3 (Figure 2b) equipped with a high-resolution RGB camera was used in this study for all image acquisition. The camera had a CMOS sensor of 12 million effective pixels (image size: 4000 × 3000). The UAV provided the accurate geographical position of each image and recorded it in the image header. The GPS/GLONASS dual-mode positioning system that the UAV was equipped with included an advanced navigation algorithm that enabled

hovering accuracy of up to $\pm 0.5$ m (vertical) and $\pm 1.5$ m (horizontal). The DJI GS Pro was used for mission planning and remote control of the UAV. UAV flight altitude was set at 55 m. The overlap rate of both forward and lateral photography was 87% (the planned flight path is shown in Figure 2c). A high overlap rate can balance the noise due to the camera sensor's pixel loss and increase the redundancy of feature points to reduce the loss in 3D reconstruction. The photography interval was 3 s, and the flight speed was 4.6 m/s.

2.1.2. Three-Dimensional PC Construction and Preprocessing

The 3D PCs were reconstructed from a set of spatially correlated images using SfM. The significant steps in this process included (1) image alignment, (2) feature point extraction and matching, (3) false match exclusion, (4) sparse PC construction, (5) dense PC construction, and (6) PC modeling. An Agisoft Photoscan (Agisoft, St. Petersburg, Russia) was used to construct PCs. The significant processes are shown in Figure 3. WGS 84 (EPSG::4326) was used as the coordinate system, consistently with all the captured images. The PC properties included coordinates (X, Y, Z), a color channel (R, G, B), and the normal vector (Nx, Ny, Nz).

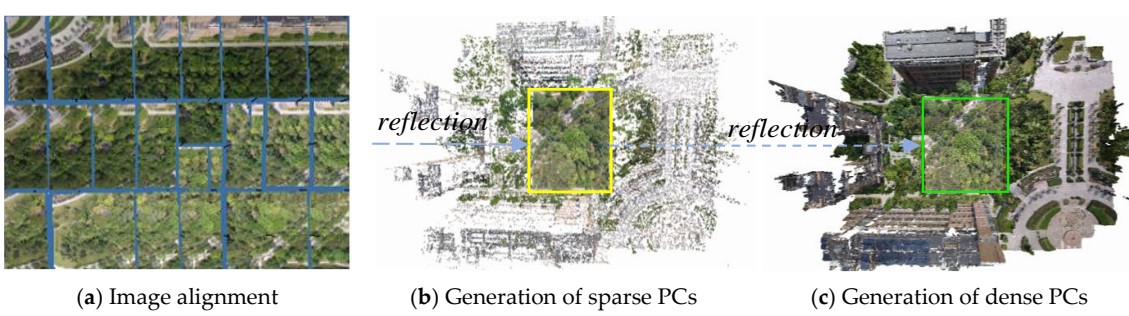

(**a**) Image alignment　　　　(**b**) Generation of sparse PCs　　　　(**c**) Generation of dense PCs

**Figure 3.** The significant processes of generating PCs via SfM. (**a**) Forest images were sequence matched and aligned, and feature points were extracted from these images. (**b**) Tie points were constructed from 2D images and formed sparse point clouds. (**c**) The sparse point clouds were filled to form dense point clouds. SfM constructs 3D PCs from 2D images and keeps the relative spatial positions of the points consistent. It is a process of "reflection".

After obtaining the PCs, some points were below the ground in clusters or scattered because of errors in the PC generation process, as shown in Figure 4. The RANSAC [35] algorithm was used to automatically fit the largest support plane to crop the subsurface points. Furthermore, the ground markers were used to segment the experimental area and remove redundancy. All processes were conducted in PyCharm using python code.

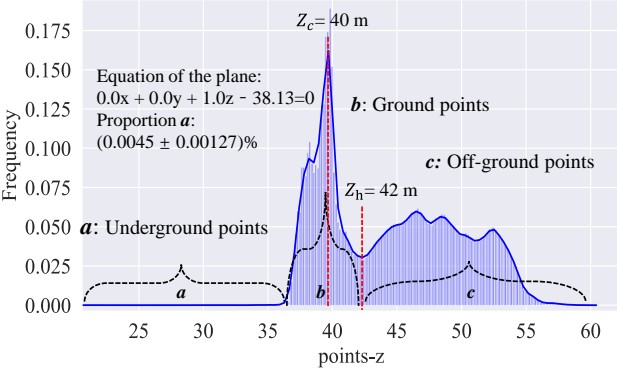

**Figure 4.** Height histogram of the experimental area's PC model. *a* indicates the underground points in the PC model. These points needed to be removed. The equation of the plane is a mathematical expression for the maximum support plane (ground), and in the experiment, a minimum of 50 points were required to satisfy this equation. $Z_c$ represents the horizontal plane, and $Z_h$ represents the highest ground points.

### 2.2. Vegetation Extraction and Filtering Noise

The vegetation needed to be extracted from the reconstructed PCs for further study. Obviously, vegetation is different from all the other objects, such as roads, cars, and buildings, in color. Three SVIs were used to extract vegetation by color.

#### 2.2.1. Spectral Vegetation Indices (SVIs)

- Excess green vegetation index (ExG)

ExG [36] is a widely used SVI for green cover extraction, which is used for crop statistics in the field by separating crops from the soil. It is calculated as follows:

$$ExG = (2G - R - B)/(R + G + B) \tag{1}$$

R, G, and B denote the red, green, and blue channels of PC color, respectively.

- Combination of green indices (COM)

Guijarro et al. [37] proposed a new vegetation index that weighted and summed four green indices: ExG, excess green minus excess red index (ExGR) [36], color index of vegetation (CIVE) [38], and vegetation index (VEG) [39], with the advantage that the weights can be adjusted for different scenarios. It is calculated as follows:

$$COM = 0.25 \times ExG + 0.3 \times ExGR + 0.33 \times CIVE + 0.12 \times VEG \tag{2}$$

where:

$$ExGR = (3G - 2.4R - B)/(R + G + B)$$

$$CIVE = 0.441R - 0.881\,G + 0.385B + 18.789745$$

$$VEG = G/(R^a \times B^{(1-a)}), \; a = 0.667.$$

- Excess green union excess red index (ExGUExR)

Vegetation changes color in autumn, as shown in Figure 5a, which means that the previous SVIs, which extracted single colors, may have missed some vegetation. A color principal component analysis was conducted on the vegetation in the test area in autumn; it was dominated by green and red. Furthermore, by testing various combined SVIs, the ExG union excess red vegetation index (ExR) [36] was proposed as a new SVI. ExR is used to extract the red color, which is calculated as:

$$ExR = (1.4R - G)/(R + G + B) \tag{3}$$

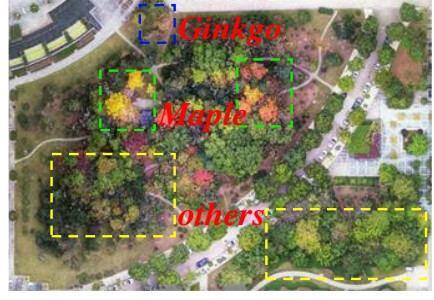

(**a**) Vegetation color variation in autumn

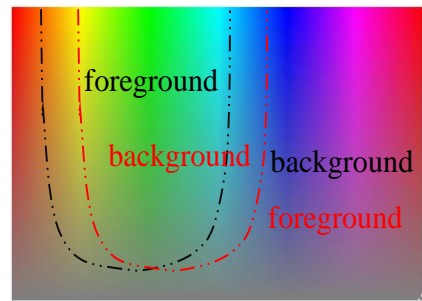

(**b**) The color gamut of ExGUExR

**Figure 5.** A demonstration of the color change of vegetation in autumn and the color gamut divided by ExGUExR. (**a**) The colors of the maple and ginkgo trees in the picture changed from green to light yellow, red, and brown. (**b**) RGB images are segmented using ExG (black lines) and ExR (red lines), and the combination of their foreground areas is the foreground area, in ExGUExR. The foreground was regarded as the interesting area.

The color range of the ExGUExR index is shown in Figure 5b. There were no brown areas in the foreground similar to the color of roads and buildings.

### 2.2.2. Threshold Selection of SVIs and Filtered PCs

Histograms were used to determine the segmentation thresholds for each index. The histograms were obtained by graying the color of the PCs with SVI. The histograms showed two peaks, representing the foreground (vegetation) and background (roads, buildings, etc.), as shown in Figure 6. The trough values of the SVIs in the histograms were chosen as the thresholds. The thresholds of the SVIs were (a) $ExG \geq 0.039$, (b) $ExR \geq 0.183$, and (c) $COM \geq 6.23$.

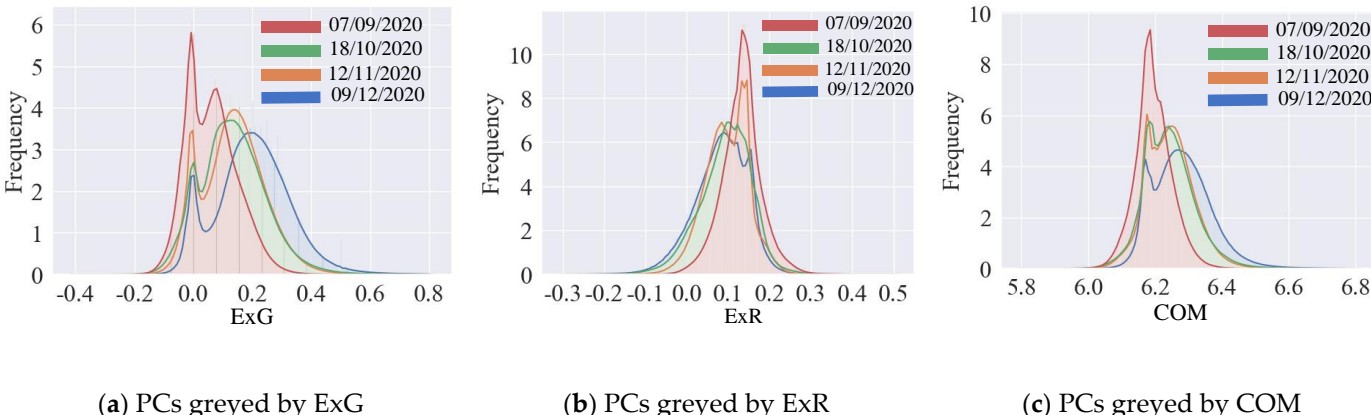

(**a**) PCs greyed by ExG $\qquad$ (**b**) PCs greyed by ExR $\qquad$ (**c**) PCs greyed by COM

**Figure 6.** Histograms were created using ExG-, COM-, and ExR-grayed PCs. The thresholds of the SVIs were (**a**) $ExG \geq 0.039$, (**b**) $ExR \geq 0.183$, and (**c**) $COM \geq 6.23$. The RGB value of each point was calculated per the equation of the relevant SVI, and the point was considered as foreground (vegetation) if the result was greater than the threshold value.

The PCs, after segmentation, contained a great deal of noise. The statistical outlier removal (SOR) method [40] was used to remove the noise points. SOR filters out points that do not have a sufficient number of neighboring points within an average radius. It requires two parameters: the number of neighboring points for mean distance estimation and the standard deviation multiplier threshold. The number of neighbors was set as 20, and thresholds were set as 2.0.

### 2.3. Separation Canopy from the Ground

The extracted PCs contained not only trees but shrubs and herbs. The trees needed to be separated using height information. As shown in Figure 4, the difference in elevation between the crest and trough was 2 m. Therefore, the elevation difference between the canopy and the ground was about 2 m (canopy height $= Z_\mathrm{h} - Z_\mathrm{c}$). The cloth simulation filter (CSF) [41] was used to separate the canopy from the ground. Previous ground filtering algorithms used mainly slope and height information to distinguish ground points from interesting points directly. The basic idea of CSF is to flip the PC and assume that a piece of cloth falls on the flipped PC's surface by gravity, and the shape of the landed cloth is taken to represent the current terrain. The separation results are shown in Figure 7b.

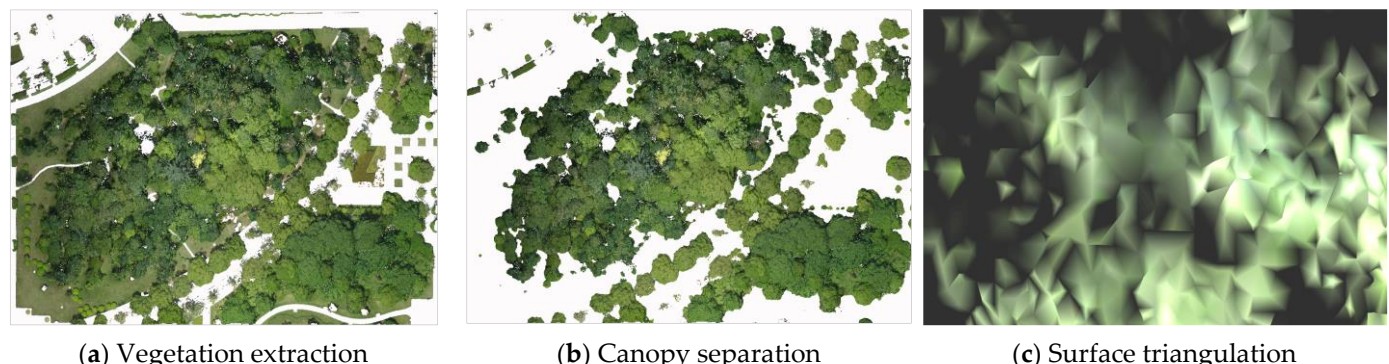

(**a**) Vegetation extraction　　　　(**b**) Canopy separation　　　　(**c**) Surface triangulation

**Figure 7.** The three major processing steps of PCs. (**a**) The vegetation was extracted by using SVIs. (**b**) Grass was separated from the canopy using CSF. (**c**) The separated canopy PCs were triangulated using Delaunay triangulation.

### 2.4. PCs Triangulation and Canopy 3D Area Measuring

After the canopy PCs were obtained, they needed to be triangulated to build topological connections between all points to calculate the 3D area (3DA). Delaunay triangulation and Poisson reconstruction are the two most used methods to establish the topology connection in PC processing. Compared with the Delaunay method, the Poisson method is used for closed structures. However, the canopy PCs were not closed. Therefore, Delaunay triangulation was better for triangulating PCs in this research. After triangulation, every three points formed a triangular mesh with the smallest possible side length. The result of this processing is shown in Figure 7c. The 3DA of the canopy was obtained by calculating the area of all of the triangular facets. Only one side of each triangular facet was counted when calculating the 3DA.

The Delaunay algorithm requires manually setting the maximum side length of the triangular facets. The patches produced by an excessive upper limit would fill the gaps between separated forest parts, while a small upper limit would cause PCs at sparse locations to be filtered out. Statistics on data sampling revealed that the best edge length limit value was between 1.5 and 2.0. The maximum relative average deviation of the measurement did not exceed 2.2%. The side length was marked as L ($L_0 = 1.5$, $L_1 = 1.6$, $L_2 = 1.7$, $L_3 = 1.8$, $L_4 = 1.9$, $L_5 = 2.0$). $S_L$ represented the PC area as the upper limit of side length is L. Therefore, the canopy area was calculated as follows:

$$\text{Canopy area} = \frac{1}{6}\sum_{L_0}^{L_5} S_L \tag{4}$$

### 2.5. Forest Canopy Coverage Area and Leaf Area Index (LAI)

Forest coverage area (CA) is an index used to characterize ecological health that is widely used in satellite-based forest monitoring [1]. PCs contain height information, which can be used to distinguish trees from green grass when calculating the forest coverage area. This can hardly be performed with satellite images. The orthogonal projection of a canopy PC to the ground represents the projected area covered by the canopy. Calculating the CA also requires Delaunay triangulation for the projected PCs. After CA is obtained, another parameter, the forest coverage rate, used to evaluate the ecological health level, can be calculated by the CA-to-land area ratio.

LAI is a widely used structural parameter of ecosystems that reflects plant foliage quantity, canopy structure changes, etc. The LAI is defined as the ratio of the sum of each tree leaf area to the tree-covered area. The traditional method of obtaining LAI has mainly been inversion [27]. This is because the adding-up of all leaf areas is impossible. However, UAV aerial photogrammetry provides a possibility for measurement of LAI, as the canopy surface area has a close relation to the sum of all leaf areas.

### 2.6. Logistic Regression

The measurement results of the canopy area (3D area and coverage area) were fitted by logistic regression. Pierre Francois Verhulst [42] first proposed the logistic function, or logistic curve, in a study on population growth patterns. The logistic function is as follows, and the fitting requires finding the parameters A1, A2, x0, and p:

$$y = \frac{A1 - A2}{1 + (\frac{x}{x0})^P} \tag{5}$$

Origin (OriginLab, Northampton, MA, USA) and a self-developed python program were used for the regression analysis of the data.

## 3. Results and Analysis

### 3.1. Canopy Area and Time-Series Analyses

The measurements of the canopy area and time-series analyses thereof are shown in Figure 8. The canopy area was represented in two ways: the 3DA (red points) and the CA (blue points). The error bars indicate the error range of the measurements, which was related to the PC triangulation parameters and used to illustrate the error range under different upper values of side length. Figure 8a–c represents the canopy areas and fitted curves from different SVIs. The $R^2$ is the coefficient of determination, which indicates the level to which the fitted logistic model explains the change in the measured values. Usually, the closer $R^2$ is to 1, the better the fitting result is.

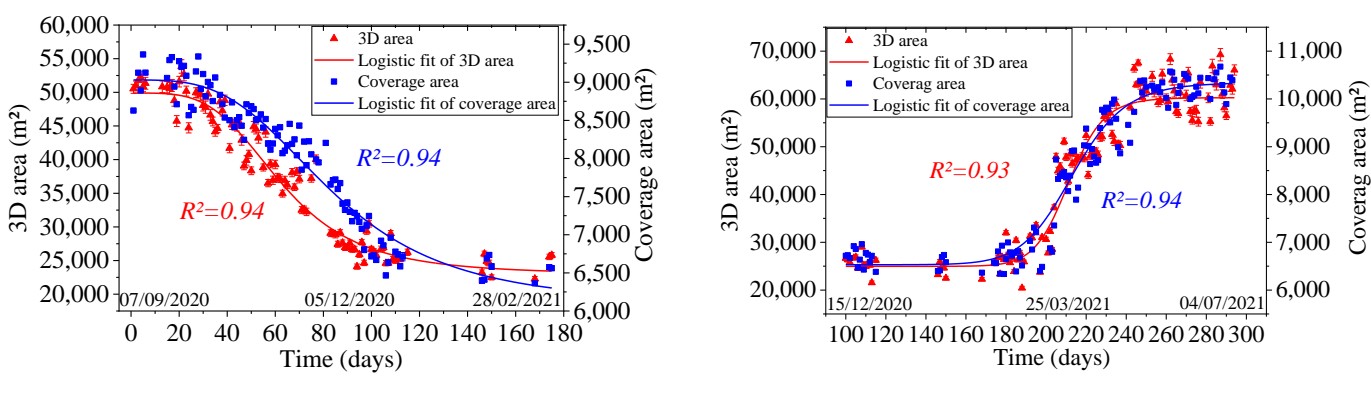

(**a1**) Defoliation period          (**a2**) Leafing period

(**a**) ExG-based measurements and fitted curves

**Figure 8.** *Cont.*

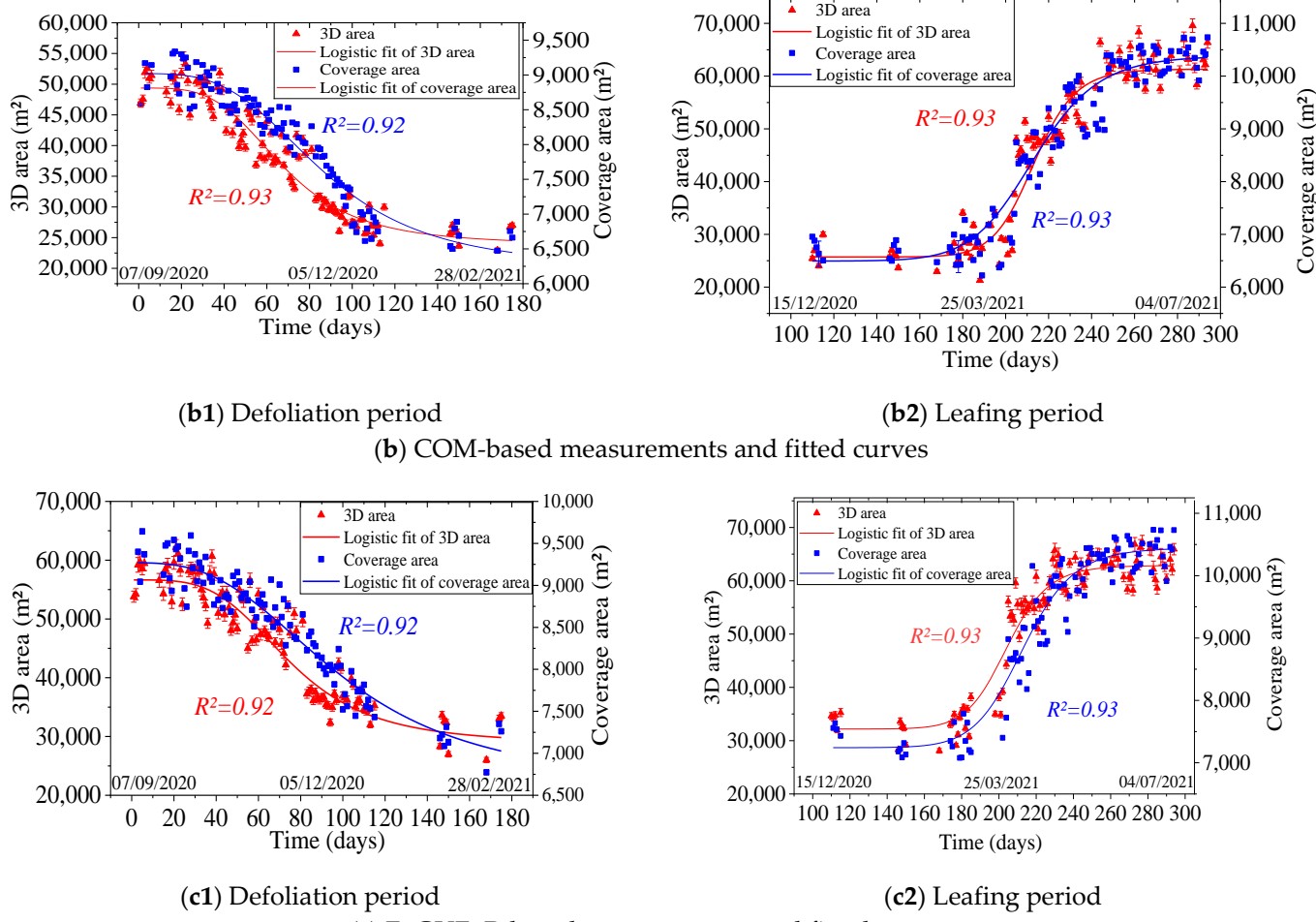

(**b1**) Defoliation period　　　　(**b2**) Leafing period

(**b**) COM-based measurements and fitted curves

(**c1**) Defoliation period　　　　(**c2**) Leafing period

(**c**) ExGUExR-based measurements and fitted curves

**Figure 8.** Three index-based measurements and time-series analyses of the 3D area and coverage area. $R^2$ is the coefficient of determination (**a**–**c**).

As shown in the graphs, the proposed method successfully measured the canopy area (3DA and CA) and successfully fitted time-series curves of canopy growth. Curves were collected for one annual growth cycle (lasting 294 days with 203 flights). According to the seasons in which the data were collected, the growth was divided into two periods: the defoliation period and the leafing period. The time-series analyses and their parameters mathematically expressed the forest growth pattern. These parameters are shown in Tables 1–3. That all values of $R^2$ were ≥0.92 showed that the fitted logistic model was well-matched with the trend of the canopy area and reflected the forest's growth pattern. This can be attributed to the high resolution of the images obtained by the UAV at an altitude of 55 m. Based on this result, it was concluded that the forest growth followed an S-shaped curve during the annual cycle, similar to the general pattern of forest growth demonstrated in [43,44].

**Table 1.** The fitted parameters of the curves obtained from the ExG-extracted PCs.

| Curve | A1 | A2 | X0 | P | R-Squared |
|---|---|---|---|---|---|
| 3DA-D | 50,123.9 ± 713.844 | 22,882.5 ± 686.61 | 62.4312 ± 1.59166 | 3.7929 ± 0.33546 | 0.94 |
| CA-D | 8997.99 ± 39.4368 | 6147.66 ± 174.537 | 83.030 ± 3.61838 | 3.52454 ± 0.35416 | 0.94 |
| 3DA-L | 24,421.3 ± 593.151 | 60,853.04 ± 1140.82 | 212.505 ± 1.34808 | 21.3015 ± 2.38037 | 0.93 |
| CA-L | 6414.93 ± 134.02 | 10,386.79 ± 79.5138 | 213.028 ± 1.45161 | 15.7965 ± 1.58301 | 0.94 |

**Notes**: 3DA-D: "Canopy 3D area in defoliation period"; others are similar.

**Table 2.** The fitted parameters of the curves obtained from the COM-extracted PCs.

| Curve | A1 | A2 | X0 | P | R-Squared |
|-------|-----|-----|-----|-----|-----------|
| 3DA-D | 49,368.4 ± 766.624 | 23,848.54 ± 885.832 | 67.6951 ± 2.10352 | 3.79503 ± 0.41044 | 0.93 |
| CA-D | 9019.31 ± 45.0857 | 6210.25 ± 213.575 | 86.4642 ± 4.66807 | 3.37483 ± 0.40016 | 0.92 |
| 3DA-L | 25,745.8 ± 643.349 | 62,077.17 ± 1193.38 | 213.54 ± 1.39402 | 22.1629 ± 2.63264 | 0.93 |
| CA-L | 6592.14 ± 126.991 | 10,379.16 ± 80.0287 | 212.507 ± 1.41324 | 16.3605 ± 1.72635 | 0.93 |

**Table 3.** The fitted parameters of the curves obtained from the ExGUExR-extracted PCs.

| Curve | A1 | A2 | X0 | P | R-Squared |
|-------|-----|-----|-----|-----|-----------|
| 3DA-D | 56,639.9 ± 841.416 | 28,754.74 ± 1138.9 | 74.9277 ± 2.51035 | 3.79381 ± 0.45909 | 0.92 |
| CA-D | 9268.64 ± 45.1712 | 6602.57 ± 262.303 | 98.6821 ± 7.22509 | 2.90367 ± 0.35385 | 0.92 |
| 3DA-L | 31,373.2 ± 580.940 | 62,344.78 ± 557.329 | 202.221 ± 1.62907 | 24.4656 ± 3.13427 | 0.93 |
| CA-L | 7250.35 ± 104.877 | 10,406.68 ± 88.3461 | 212.867 ± 1.46264 | 16.2736 ± 1.99931 | 0.93 |

*3.2. Canopy Growth Curve and Growth Rate Curve Analysis*

Based on the above analysis, the time-series analysis curves of the canopy area were considered the growth curves of the canopy. To further obtain growth rate curves, the first-order derivatives of the growth curves were calculated. The growth curves and growth rate curves are shown in Figure 9. Parts I and II represent the growth rate curves of 3DA and CA, respectively.

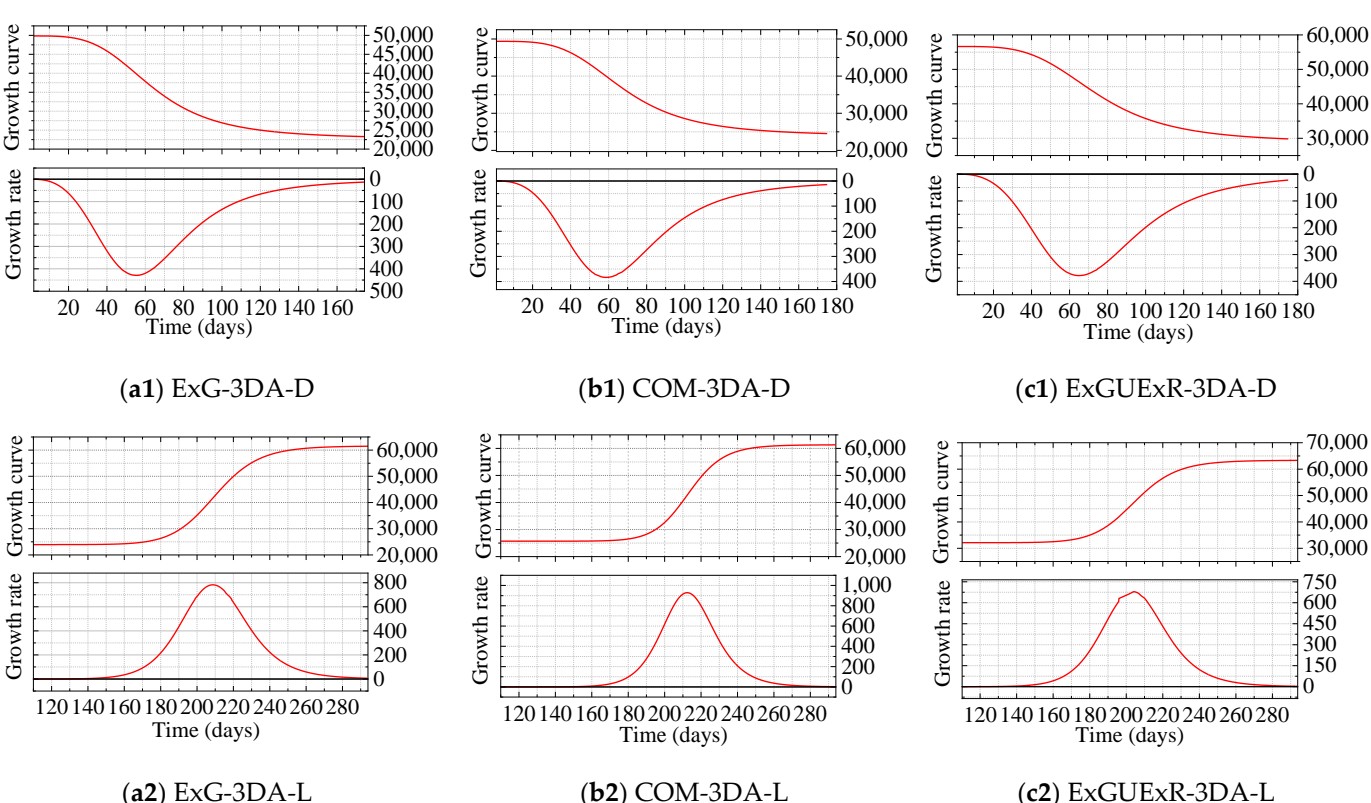

(**a1**) ExG-3DA-D  (**b1**) COM-3DA-D  (**c1**) ExGUExR-3DA-D

(**a2**) ExG-3DA-L  (**b2**) COM-3DA-L  (**c2**) ExGUExR-3DA-L

**Figure 9.** *Cont.*

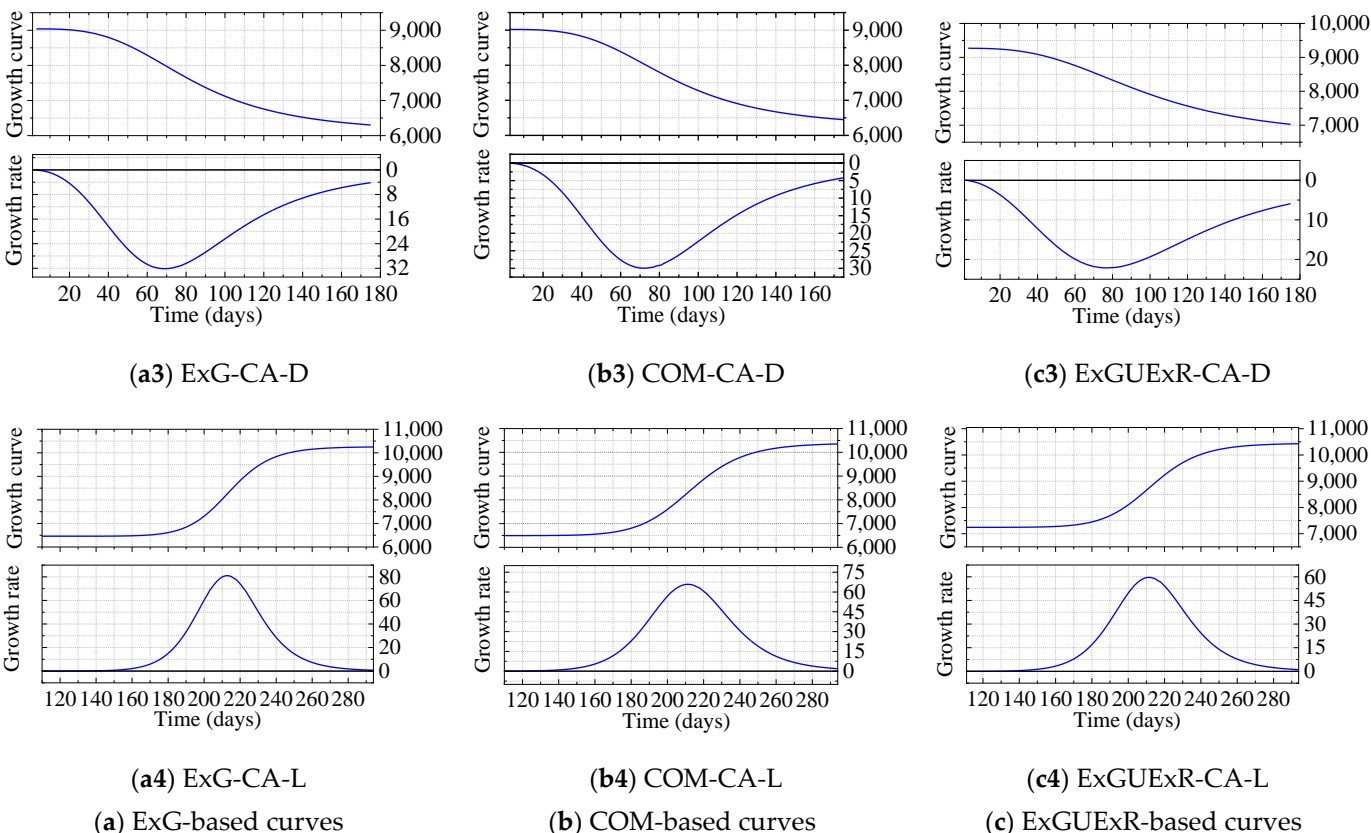

(**a3**) ExG-CA-D　　　　(**b3**) COM-CA-D　　　　(**c3**) ExGUExR-CA-D

(**a4**) ExG-CA-L　　　　(**b4**) COM-CA-L　　　　(**c4**) ExGUExR-CA-L

(**a**) ExG-based curves　　(**b**) COM-based curves　　(**c**) ExGUExR-based curves

**Figure 9.** Growth curves and growth rate curves of the canopy. The rate curve is the first-order derivative curve of the growth curve, during the defoliation period, this curve indicates the rate of canopy area reduction, and during the leaf-bearing period, it indicates the rate of canopy area increase. Column (**a**), ExG-based growth and rate curves. Column (**b**), COM-based growth and rate curves. Column (**c**), ExGUExR-based growth and rate curves. ExG-3DA-D: "ExG index-based canopy growth and rate curves in defoliation period". Others are similar.

### 3.2.1. General Growth Analysis Based on the Growth Curves and Rate Curves

The growth curves showed that the forest canopy in the experimental area was stable in the summer months and gradually decreased in October. Since there were both deciduous and evergreen trees in the experimental area, the growth curves endured a long decline process. In the spring, the growth curves increased rapidly in a short time and began to stabilize at the end of the spring. The growth rate of the canopy area obeyed a "slow–fast–slow" pattern; that is, the rate of decrease (or increase) in the canopy area reached a peak first and then gradually decreased. The canopy growth time was shorter than the defoliation time, and the peak growth rate was greater than the peak defoliation rate. This was perhaps due to the warm climate and sufficient rainfall in spring, which are favorable for tree cell reproduction [45–47].

### 3.2.2. Comparison of SVI-Extracted Vegetation and Their Curves

In this research, three SVIs were used to extract the forest canopy. The extraction results thereof are shown in Figure 10. For better display purposes, the canopy was not extracted separately. As seen in the figures, all indices removed the roads and buildings clearly, but the extracted vegetation had different levels of integrity. In summer, the vegetation extracted by the three indices was similar; a closer look reveals that the ExGUExR-index-extracted vegetation had fewer holes and that, in fact, the void region was almost all green. In autumn, the vegetation extracted by the three SVIs was significantly different. The ExG and COM indices extracted only green and a few yellow areas, while the ExGUExR index extracted almost all vegetation.

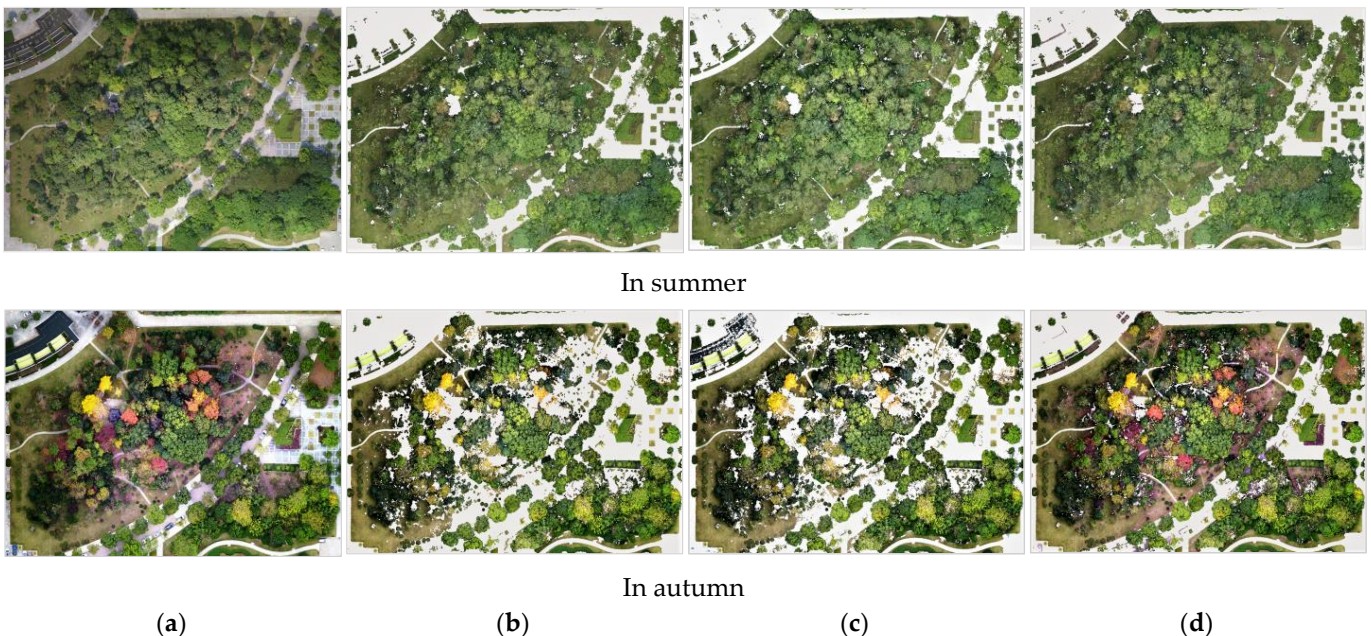

In summer

In autumn

(**a**)                    (**b**)                    (**c**)                    (**d**)

**Figure 10.** Comparison of the vegetation extracted by the ExG, COM, and ExGUExR indices at different growth stages (summer and autumn). (**a**) Raw PCs of the experimental area; (**b**) vegetation extracted by ExG; (**c**) vegetation extracted by COM; (**d**) vegetation extracted by ExGUExR.

Similar conclusions could be obtained from Figure 10 as from Figures 8 and 9; i.e., on the same day, the measured values of ExG and COM were close to each other. The values measured by ExGUExR were the largest. Therefore, the logistic regression curves of ExG and COM were very close in morphology and parameters. These indices corresponded to two physical connotations. The ExG and COM indices extracted mainly the green component in the canopy, while the ExGUExR index extracted almost the whole canopy. However, even in autumn, the vast majority of the areas in the extracted canopy (in Figure 10, the complete vegetation area is shown without canopy separation) remained green, so overall, the logistic regression curves of ExGUExR were close to those of ExG and COM. According to this and the relevant results: (1) during the same period, the canopy and the green regions in the canopy had similar growth patterns, and the values of the logistic function parameters are also close. However, in the defoliation period, the canopy area reached the peak rate later than the green regions in the canopy; (2) in the leaf-bearing period, the canopy area reached the peak rate before the green area in the canopy.

### 3.2.3. Comparison between 3DA and CA Curves

The 3DA and CA curves are shown in Figure 9. They characterized canopy growth differently. The 3DA rate curves reached their peak about 10 days earlier than the CA rate curves during the defoliation period. During the defoliation period, the canopy changed from dense to sparse, and the leaves were heavily shaded against each other such that the lateral area change could not be observed in the orthograph. Therefore, the CA produced only a tiny reduction when the 3DA was heavily reduced. However, there was almost no shading during the leaf-bearing period, so the growth trends in 2D and 3D were relatively uniform.

In Figure 11, the correlations between 3DA and CA are shown, $0.93 \leq$ *Pearson's r* $\leq 0.95$ ($0.87 \leq$ coefficient of determination $R^2 \leq 0.9$), which quantitatively illustrates the level of their consistency in characterizing canopy growth. The basic trends of CA and 3DA were consistent, but CA described only the 2D expansion (and decay) of the canopy in the plane. In summary, CA, as a 2D analysis method, has the obvious drawback of missing a dimension. Therefore, though the CA of the canopy can be obtained more easily by PCs, using CA to characterize canopy growth may not be the best choice.

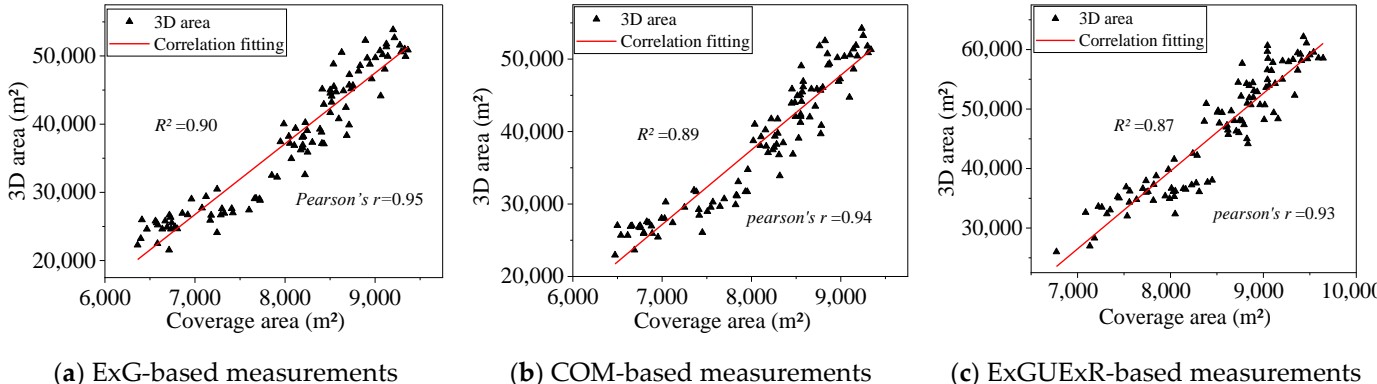

(**a**) ExG-based measurements  (**b**) COM-based measurements  (**c**) ExGUExR-based measurements

**Figure 11.** Correlation analysis of 3DA and CA obtained by linear fitting. $R^2$ is the coefficient of determination.

### 3.2.4. Calculation of Forest Coverage Rate and LAI

As mentioned in Section 2.5, forest coverage rate and LAI can also be obtained from PCs. They are calculated as follows:

$$\text{Forest coverage rate} = \frac{\text{Forest coverage area } (\text{m}^2)}{\text{Land area } (\text{m}^2)} \tag{6}$$

$$\text{LAI} = \frac{\text{Total leaves area } (\text{m}^2)}{\text{Land area } (\text{m}^2)} \tag{7}$$

As an example to illustrate this approach, in summer, the forest coverage area of the experimental site was 10,500 m$^2$, and the experimental area was 16,692 m$^2$; thus, the forest coverage rate is 62.9%. The canopy 3D area was about 60,000 m$^2$, so the LAI value was about 3.59. In the autumn, the most credible data were based on the results of the ExGUExR index; otherwise, the results skewed small. Based on this result and reference to biological and environmental evaluation indicators, it is possible to estimate the level of ecological and vegetation health. In the case of the experimental area, the forest coverage rate exceeded the average in China [48], and the forest growth was healthy [49].

## 4. Discussion
### 4.1. Error Analysis
#### 4.1.1. Error Source Analysis

In the experiment, the first source of error lay in the geographic coordinates of images. In open areas, the positioning error of the UAV was less than the limit values of ±0.5 m in the vertical and ±1.5 m in the horizontal direction. While since the experiment site was on the campus, the surrounding buildings influenced the accuracy of positioning. The second source of error was camera position resolving error during the PC generation. However, this error was relative to the geographic coordinates in the UAV images. These two sources of errors eventually accumulated in the generated PCs. Since we had no available means by which to obtain the true value of the canopy area of this forest, the system error level was illustrated by comparing a control length in the experimental area with the size of the same object in PCs.

As shown in Table 4, the length and width of the experimental area in the PCs were sampled for 10 days. According to the results, the systematic error did not exceed 0.8%.

**Table 4.** The length and width of the test area were measured from the PCs.

| | True Value | | | | Measured Value | | | | | | | Average Error |
|---|---|---|---|---|---|---|---|---|---|---|---|---|
| Length (m) | 154.9 | 156 | 156 | 156 | 156 | 157 | 155 | 157 | 156 | 156 | 156 | 1.2 |
| Width (m) | 106.2 | 107 | 107 | 107 | 107 | 106 | 107 | 108 | 107 | 107 | 106 | 0.7 |

### 4.1.2. Analysis of Random Errors in Measurements

The residuals (RESID, error between measurements and fitted values) were used here to demonstrate the level of random error in the 3DA measurements, as shown in Figure 12. In fact, even without considering the cost and deployment effectiveness advantages, the measurement accuracy based on aerial photogrammetry was encouraging compared with that based on satellites, the resolution of which is limited in small areas and subject to atmospheric influences. Solar illumination was the primary source of random errors (take the fitted value as the truth value) in the measurements. In the experiment, the weather data were divided into strong illumination and normal illumination according to the light intensity. The statistics revealed that the data distribution was more discrete under strong illumination conditions. In Figure 12, residual values extremely far from the 0-scale line were obtained most often under strong illumination. The normal illumination measurements were generally close to 0.

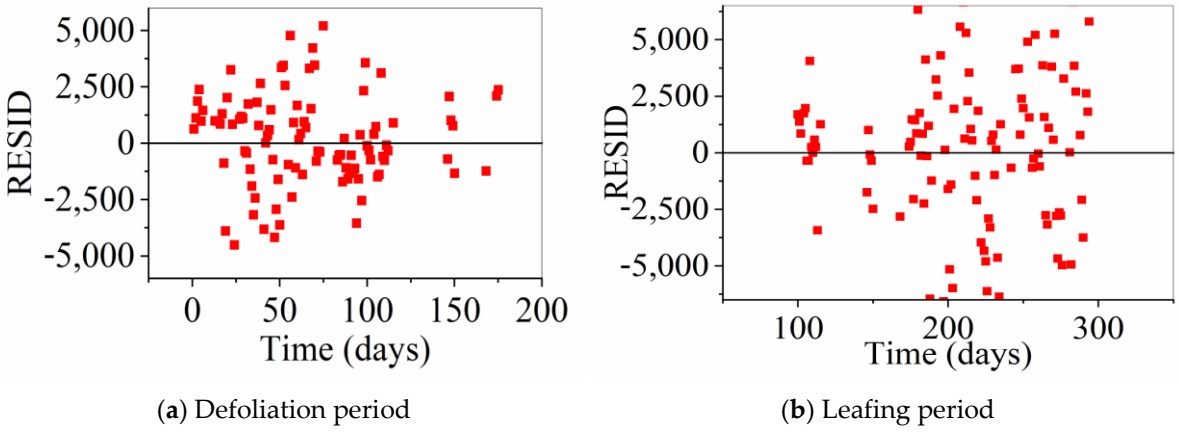

(**a**) Defoliation period  (**b**) Leafing period

**Figure 12.** Regular residuals of 3DA measurements.

Table 5 shows the PC reconstruction parameters under intense illumination and normal illumination, the latter of which were greater than the former. This indicates that the conversion efficiency of transforming the 3D spatial structure model from the 2D images was higher, and the PC quality was better, under normal illumination.

**Table 5.** PC quality statistics under strong and normal illumination conditions.

| Illumination Grade | Sample Quantity | Average Tie Points | Average Dense PC Points | Average Point Density |
|---|---|---|---|---|
| intense | 55 | 125,992 | 12,789,007 | 137 |
| normal | 56 | 132,434 | 13,441,000 | 143 |

Under intense illumination, shadows were generated locally on the forest surface because of leaf shading. The grayscale variation in the shaded area was too significant, making it difficult to extract information from this area and generating local deficiencies in the reconstruction. In the step of sparse PC generation, the relative position estimation between images was performed using SIFT feature points. Thus, the possible reasons for the missing reconstruction data were supposed as follows: (1) uneven illumination may have led to partial corner point miss detection and false detection and then reduced the accuracy of corner point detection under intense illumination; (2) the uneven illumination may have reduced the accuracy of feature extraction, such as in the case of edge corner points, which would have reduced the calibration accuracy.

In summary, UAV aerial photogrammetry, as a method with significant advantages of excellent deployment efficiency, high ground resolution, and accessibility of data, could effectively be used for daily monitoring and description of the growth pattern of a forest. With higher budgets, using UAVs carrying RTK positioning equipment or airborne

LiDAR may be better to reduce errors. UAVs with RTK can further reduce system errors, which may benefit the quality of canopy PCs. UAVs equipped with LiDAR can generate higher-accuracy PCs, but their colors depend on the fusion with RGB images. Furthermore, small LiDAR sensors suitable for UAV applications are expensive and are limited to a few manufacturers. They are far more costly than small-scale UAVs with digital cameras [50]. Larger UAVs, such as fixed-wing drones and oil-powered helicopters, can extend flight duration and enlarge coverage so as to promote and validate the time-series forest monitoring method in much larger experiments.

### 4.2. Comparison with Other Studies

In some similar studies, researchers measured tree height or crown spread to characterize tree growth using UAVs and SfM [51,52]. However, tree height measurements are usually taken in areas where trees are sparse. In a study by Mlambo et al. [53], the researchers evaluated SfM horizontal and vertical accuracy for measuring the height of individual trees. The results showed that, at Dryden, poor correlation was observed between SfM tree heights and ground-measured heights ($R^2 = 0.19$). Obviously, UAVs cannot capture the ground in dense forests because of canopy shading. Therefore, in the PCs generated by SfM, tree height is difficult to measure accurately, because there is no ground reference. This view was also confirmed in a study by Kameyama et al. [33]. In 16 conditions, tree height and crown were undecipherable in their experiment. In a study by Miller et al. [54], the researchers used multiview stereophotogrammetry and SfM (SfM–MVS) to measure individual tree height, crown spread, crown depth, stem diameter, and volume. The results showed that, apart from height and crown depth, all modeled variables had a negative bias, suggesting that SfM–MVS tended to underestimate the size of trees. However, tree height, crown spread, crown depth, and stem diameter can represent the growth of trees only in a particular direction/plane. Canopy volume calculation requires the model structure to be closed, which is easy to achieve for individual trees, but in forest PCs, there are gaps in the crowns, and the whole forest canopy is an unclosed surface.

The method proposed herein can be used for canopy 3D area measurements of individual trees and entire forests (careful segmentation would be required to measure a single tree canopy in a dense forest). Moreover, 3DA was shown to be a 3D indicator that can comprehensively characterize tree growth.

### 4.3. The Applicability of the Proposed Method in Natural Forests

In this study, a small forest on campus was chosen as the test subject. Since it is difficult for drones to photograph forests at the city level, discussion of the applicability of the proposed method should focus on scenarios such as forest parks or nursery gardens. Typically, forest parks do not exceed 20 square kilometers in China, so they can be photographed by dividing the area or using large, long-endurance drones. The flight altitude of drones can be flexibly adjusted. The natural forest airspace is clean and open. Thus, the flight altitude can be set lower than the 55 m set in this experiment to obtain higher-resolution ground images if there are no concerns about the battery life of the UAV. Under natural conditions, the color contrast between soil and trees is obvious; usually, the ground is brownish, and the vegetation is green. Therefore, the use of SVIs may hardly be restricted in most situations. Natural forest parks have diverse topography. Although the complex terrains have a negative impact on canopy extraction, successful application in numerous studies has proven that CSF may have utility in most terrains [55–57]. The other PC processing mainly involves hunting for suitable parameters, which can be obtained by a few attempts or statistical analysis. In summary, according to analysis of the experiment and other relevant studies available [58–60], we believe that the proposed method has applicability in forestry.

## 5. Conclusions

In this paper, we proposed and tested a method for calculating the 3D area of a forest canopy to characterize annual canopy growth in forests. The proposed method was based on UAV remote sensing and PC processing. In the study, (1) three SVIs (ExG, COM, ExGUExR) were used to extract the vegetation in PCs based on the colors. The results showed that ExG and COM extracted the green areas in the canopy, while ExGUExR extracted the complete canopy; (2) a more convenient method of obtaining CA from a canopy PC was demonstrated, and the differences between 3DA and CA curves describing canopy growth were compared, with the conclusion that the canopy growth described by 3DA was more accurate; (3) based on the basic definition of LAI and the knowledge that leaves cluster, forming canopies, a novel method of estimating LAI was proposed and successfully verified. A new method of calculating forest coverage rate was also demonstrated.

Both the systematic and random errors are discussed, and in general, the accuracy of the measured values met expectations. Compared with methods in other studies, the proposed method can depict forest dynamics in a 3D space, which could have multiple applications in forestry.

In summary, with the main advantages of efficient deployment, high ground resolution, and easy data acquisition, the UAV aerial photogrammetry method was successfully used for daily monitoring and annual growth trend description of the forest in this experiment. The potential use of this study is to provide new production management methods for forestry in growth monitoring, growth trend description, etc.

**Author Contributions:** Conceptualization, W.Z., C.Z. and Y.Z.; methodology, W.Z., F.G. and N.J.; software, validation, formal analysis, and writing—original draft preparation, W.Z.; resources, project administration, funding acquisition, supervision, Y.Z.; writing—review and editing, C.Z. and Y.Z. All authors have read and agreed to the published version of the manuscript.

**Funding:** This research was funded by the National Natural Science Foundation of China (NSFC), grant number: 61905219.

**Data Availability Statement:** Not applicable.

**Conflicts of Interest:** The authors declare no conflict of interest.

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
