# Peer review of "High-Temporal-Resolution Forest Growth Monitoring Based on Segmented 3D Canopy Surface from UAV Aerial Photogrammetry"

_drones, doi:10.3390/drones6070158_

Round 1
Reviewer 1 Report
The authors of the text deal with the issue of forest monitoring. The monitoring was mainly carried out from the evaluation of images or orthophotos taken from aircraft or satellites. The required 3D imaging of forest stands from drones at low altitudes has not been studied in significant depth so far.
Recently, the availability of high-resolution 3D data has made it possible to obtain accurate information on the size of the tree canopy, making the research area of 3D monitoring of canopy condition and growth a hot topic.
In the text of this paper, a forest growth model has been studied by evaluating a point cloud of the canopy reconstructed from an image obtained from UAV aerial photogrammetry at daily intervals over a period of one year. The growth curves were obtained by evaluating the canopy area calculated from the triangulated 3D mesh. Methods for determining canopy area (CA), forest cover ratio and leaf area index (LAI) were proposed and tested. Three vegetation indices were used for tree segmentation: excess green index (ExG), combination of green indices (COM) and excess red bundle index (ExGUExR). The results of analysis and comparison show that the vegetation areas extracted by ExGUExR are more complete than the other 2 indices. Furthermore, that logistic fitting of 3DA and CA provided S-shaped growth curves with all correlations. Finally, the 3DA curves appeared to represent the growth dependence more accurately than the CA curves.
The study discussed the measurement errors and considered the possible use of the growth curves for forest management, for example, the level of carbon and oxygen exchange in the atmosphere around the stand. The UAV aerial photogrammetry method was successfully used for daily monitoring and description of the annual growth trend.
The authors provide an elaboration of insights into the potential use and comparison of the sensed data for forest management and quality purposes. In part, they bring new insights from image processing to evaluate the resulting growth parameters of individual trees. The text is written on the basis of scientific procedures and is supplemented with the necessary references and image documentation.
I recommend checking the English, for example in the title "4.4 The discussion of the mehod applicability in natural forests ", line 463 there seems to be a transcription error - "method" instead of "mehod" , etc.
The formal notation and description of the pictorial documentation is of a sufficient standard. In Fig.6 the values and data on the x and y axes are not legible, in Fig.4 the figure axis labels are in small print and harder to read. When evaluating the data and the analyzed/calculated results in, for example, Tab.1, Tab.2, Tab.3, the number of valid digits of the variable is incompatible when evaluating the uncertainty of the variable. Please correct this.
Author Response
Dear Reviewer, thanks for the careful review of our manuscript! We have carefully considered all your suggestions and have made corresponding revisions to the manuscript. Please review it in the response file.Hope our presentation meets your expectation。

Reviewer 2 Report
Dear Authors,
I personally like the content of your manuscript but the presentation, especially the use of the English language needs to be thoroughly reviewed. There is speculation in the results that needs to be supported by citations.
As a rule of thumb in a scientific paper you should not use apostrophe ( ' ). It is used repeatedly in the paper and it needs to be modified and another one is never to start a sentence with a number. You can write 'Three' but not '3'.
There are several sentences that I could not understand clearly (line 66-69. 234-236, 378-380, 500-501).
The objectives are quite repetitive and more clarity is needed among them.
There are some sentences that could be erased for a better flow of the paper:
Line 11, 250-252, 334, 476-478 (just leave 'the proposed methodology has a potential applicability in forestry', start line 491 from ... Growth curves..., 500-501).
Also please be careful when you use sentences like 370 ...revealed the abscisic acid action, here you need a reference.
There are many small English corrections and I could correct them all but it is better if you have someone checked the paper thoroughly and can they be quickly corrected. Once it is checked I will be more than glad to review it again.
Author Response

(The authors gave the same response as above.)

Reviewer 3 Report
The article is very interesting and comprehensive and addresses a very topical issue. My comments are rather formal.
A number of abbreviations are not explained until much further down in the text (e.g. 3DA, PC are mentioned in the Introduction but not explained until the Methodology).
I would personally move the data processing workflow after the description of the site and the equipment used.
Row 116-117 - To make the sunshine condition consistent among all acquisitions, all photo-shooting flights were taken between 12:00 and 14:00. This is not a true statement as the sunshine condition were different due to the position of the sun.
Figure 7 c) - Surface triangulation is not visible, please try to change color palette.
I have no major comments on the actual methodology and processing of the results. I just don't know why Ground Control Points were not used to refine the position, it would not then be necessary to deal with the accuracy in the extent of the area.
There are a few typos in the text (e.g. in Figure 1 - Time series anlysis....analysis, Figure 2 - DJI pantom 3.....Phantom).
After correcting formal errors and explaining abbreviations in the Introduction, I recommend it for publication.
Author Response

(The authors gave the same response as above.)

Round 2
Reviewer 2 Report
Now that the English proof-reading has been performed, it is much clearer to understand its content. The data collection is very intense and it is a new use of the drones in forest research and a bigger scale than a still camera.this approach in the suitable conditions can be replicated in the future.
I have some minor comments below: Line 10: … forest monitoring has been mainly performed… Line 11: In recent years… Line 67: erase ‘to’. Line 123: is there any particular reason why you set up the flying altitude at 55 m? Line 235:…ground is the projected area covered by the canopy… Line 255 and 256: erase ‘graphs’ Line 295-297: this is an important finding that could be match with available citations about this topic. Line 446: do you mean ‘parameters’ Line 464-466: … in a 3D space, that can have multiple applications in forestry. ( erase the sentence in between)Author Response
Dear Reviewer, we deeply appreciate your advice and help in improving our manuscript. We have carefully considered all your suggestions and have made revisions accordingly in the manuscript. Please see it in the response. Thanks!
